# Analytical Validation of a Genomic Newborn Screening Workflow

**DOI:** 10.3390/ijns11040091

**Published:** 2025-10-10

**Authors:** Kristine Hovhannesyan, Laura Helou, Benoit Charloteaux, Valerie Jacquemin, Flavia Piazzon, Myriam Mni, Charlotte Flohimont, Corinne Fasquelle, Davood Mashhadizadeh, Tamara Dangouloff, Vincent Bours, Laurent Servais, Leonor Palmeira, François Boemer

**Affiliations:** 1Biochemical Genetics Laboratory, CHU of Liege, University of Liege, Avenue de l’Hôpital 1, 4000 Liege, Belgium; vjacquemin@chuliege.be (V.J.); myriam.mni@chuliege.be (M.M.); cflohimont@chuliege.be (C.F.); f.boemer@chuliege.be (F.B.); 2Department of Human Genetics, CHU of Liege, University of Liege, Avenue de l’Hôpital 1, 4000 Liege, Belgium; laura.helou@chuliege.be (L.H.); benoit.charloteaux@chuliege.be (B.C.); corinne.fasquelle@chuliege.be (C.F.); vbours@chuliege.be (V.B.); lpalmeira@chuliege.be (L.P.); 3Division of Child Neurology, Department of Pediatrics, Reference Center for Neuromuscular Diseases, CHU of Liege, University of Liege, Bd du Douzième de Ligne 1, 4000 Liege, Belgium; flapiazzon@gmail.com (F.P.); davood.mashhadizadeh@guest.uliege.be (D.M.); tamara.dangouloff@uliege.be (T.D.); 4MDUK Oxford Neuromuscular Centre, NIHR Oxford Biomedical Research Centre, University of Oxford, John Radcliffe, Headington, Oxford OX3 9DU, UK; laurent.servais@paediatrics.ox.ac.uk

**Keywords:** newborn screening, genomic, BabyDetect, dried blood spot, next generation sequencing, analytical validation

## Abstract

Newborn screening (NBS) has evolved significantly since its inception, yet many treatable rare diseases remain unscreened due to technical limitations. The BabyDetect study used gene panel sequencing to expand NBS to treatable conditions not covered by conventional biochemical screening. We present here the analytical validation of this workflow, assessing sensitivity, precision, and reproducibility using dried blood spots from newborns. We implemented strict quality control thresholds for sequencing, coverage, and contamination, ensuring high reliability. Longitudinal monitoring confirmed consistent performance across more than 5900 samples. Automation of DNA extraction improved scalability, and a panel redesign enhanced the coverage and selection of targeted regions. By focusing on known pathogenic/likely pathogenic variants, we minimized false positives and maintained clinical actionability. Our findings demonstrate that gene panel sequencing-based NBS is feasible, accurate, and scalable, addressing critical gaps in current screening programs.

## 1. Introduction

Newborn screening (NBS) has come a long way since its first introduction to the public health system back in the 1960s [1]. The advent of new technologies has paved the path for progressive inclusion of more metabolic and endocrine disorders, resulting in the saving of thousands of children from severe disability and/or early death [2]. The inclusion of a disease in NBS is driven by the criteria of Wilson and Jungner, that include the existence of an efficient treatment. Moreover, early screening and diagnosis of diseases are of primary importance to obtain the best effect of such treatments and to accelerate new drug development. However, today, a set of rare diseases that completely match the criteria of Wilson and Jungner are not screened at birth. Among those diseases are congenital myasthenic syndromes [3], neurotransmitter diseases [4], pyridoxine- and pyridoxal-dependent epilepsy [5], Brown–Vialetto–Von Laere syndrome [6], and Wilson disease [7]. Even though there is substantial pre-clinical and clinical evidence that early treatment is much more effective than late treatment, and that pre-symptomatically treated patients have a much better prognosis than patients treated after the onset of symptoms, spinal muscular atrophy (SMA) is only recently being screened at birth in Europe. The case of severe combined immunodeficiency (SCID) constitutes another perfect example [8].

In many countries, comprehensive NBS programs have not yet been established as standard practices in neonatal care. The main reasons for the absence of NBS for eligible diseases are technical. All NBSs today are implemented as metabolic-based and for some exceptions—DNA-based. As a consequence, diseases without a well-identified biomarker or metabolite, measurable in excess or in default, are generally not screened.

Another issue is the time needed in several countries to introduce new diseases one by one in the official programs. Spinal muscular atrophy constitutes an excellent example. Indeed, this condition that now constitutes the paradigm of a “must-be-screened” has been detected at birth since 2018 in Belgium [9] or Germany [10], but not yet in the UK [11].

Pilot initiatives in genomic newborn sequencing have been blossoming worldwide over the past few years [12,13,14,15,16,17,18]. We have initiated a 3-year research project, BabyDetect (ClinicalTrials.gov: NCT05687474; www.babydetect.com, accessed on 2 June 2025), using targeted panel sequencing (tNGS). In this paper, we describe the workflow of BabyDetect together with analytical validation. We highlight the importance of defining and implementing the quality parameters for longitudinal monitoring of the routine genomic NBS workflow. We demonstrate the potential and challenges of panel-based screening to be implemented as an NBS program with the capacity to screen more diseases.

In this first implementation, no copy-number or structural variant (CNV/SV) analysis was performed due to lack of sufficient positive controls for validation. The detection of CNVs and other SVs is an essential component of comprehensive genetic screening, as such variants can indeed be pathogenic and disease-causing. Here, we present a pragmatic plan to enable CNV detection and analysis in future releases while preserving screening-grade performance.

## 2. Materials and Methods

### 2.1. Samples

Parental-informed decision was taken before collection of newborns’ samples on dedicated filter paper cards (LaCAR MDx, Liège, Belgium) designed for the BabyDetect project [14,19]. These cards were used to keep research samples separate from the routine NBS workflow and to streamline logistics and traceability. The project was approved by the Institutional Review Board (CHU Liege ethics committee (n° 2021/239)) and complied with the WMA Declaration of Helsinki.

### 2.2. Design of Validation Plates

Two microtiter plates of 88 samples were designed for method validation. Plates contained DNA extracted from newborns’ dried blood spots (DBSs) and the DNA extracted from whole blood. The outline of the validation plates is presented in Appendix A. The following samples were used for validation purposes:Eight positive newborn samples (NBPOS-1 to NBPOS-8, 3-plex of each), with pathogenic (P) or likely pathogenic (LP) variants in *PAH*, *ACADM*, *MMUT*, *G6PD*, *CFTR*, *DDC* genes, confirmed with screening and diagnostic methods.Eight negative newborn samples (NBNEG-1 to NBNEG-8, 3-plex of each).Four negative adult samples (ADNEG-1 to ADNEG-4, 4-plex of each). These are adult whole blood samples with no reported conditions.Same four negative adult samples spotted on DBS (DBS-ADNEG-1 to DBS-ADNEG-4, 4-plex each).HG002-NA24385 Genome in a Bottle (GIAB) reference DNA (https://www.nist.gov/programs-projects/genome-bottle, accessed on 22 September 2025, Coriell Institute, Camden, NJ, USA), 8-plex.

### 2.3. DNA Extraction

DNA was extracted from DBS and whole blood manually using the QIAamp DNA Investigator Kit (Qiagen, Hilden, Germany) according to the manufacturer’s instructions with modifications for plate-based extraction. Manual extraction was used for initial validation samples as well as during the first year of screening.

After one year of screening, in order to ensure scalability of the population-based NBS and to improve turnaround time (TAT), automated extraction was implemented using the QIAsymphony SP instrument (Qiagen, Hilden, Germany) with the QIAsymphony DNA Investigator Kit (Qiagen, Hilden, Germany). Both manual and automated methods were validated and compared.

DNA yield was quantified using a Qubit fluorometer (Thermo Fisher Scientific, Singapore), and DNA quality and fragment size were assessed by agarose gel electrophoresis and Agilent fragment analysis (Appendix A).

### 2.4. tNGS Panel Design and Sequencing

Eligible diseases were selected by performing a systematic literature review and having discussions with local medical experts. Gene inclusion criteria were considered before selecting genes of interest: early onset of disease (before age of 5), severe disease/disability if not treated, available treatment, benefit from treatment, pathogenic mutation, phenotype–genotype correlation, and standard of care (Appendix A).

Firstly, a custom target panel covering 359 genes for 126 diseases was designed (panel-v1) and it was later expanded to reach 405 genes for 165 diseases (panel-v2) (Appendix A) [14]. Twist Bioscience technology (San Francisco, CA, USA) was used for library preparation and the high-performing probes were selected for target enrichment.

For panel-v1, probes were designed to capture all coding regions of genes of interest, including 3′UTRs and 5′UTRs. For 20 complex genes, introns were also targeted. Altogether, targets of interest (TOIs) in panel-v1 encompassed 1.6M bases of genomic DNA.

To improve the quality and performance of the BabyDetect panel, we designed a second version of the panel (panel-v2). The original design of panel-v1 was modified to focus on only the coding regions and intron–exon boundaries (~50 base pairs from the intronic borders) of selected genes. Deep intronic variants, promoter and UTRs, and homopolymeric regions were not targeted. With panel-v2 approximately 1.5 Mb were targeted for capture and sequencing. The primary technical rationale for redesigning panel-v2 was to improve on-target capture efficiency. In panel-v1, intronic regions as well as 3′ and 5′UTRs from 20 genes had been included in the target of interest (TOI), but these noncoding regions were excluded in panel-v2 to enhance performance. In addition, the redesign enabled the incorporation of newly curated genes of interest identified during a second round of gene curation (Appendix A).

Study samples were sequenced using Illumina technologies. Three validation runs were realized: two independent runs with 2 × 100 bp reads on NovaSeq 6000 (20040719, NovaSeq 6000 SP Reagent Kit v1.5, 200 cycles, San Diego, CA, USA) and one run with 2 × 75 bp on a High Output NextSeq 500/550 systems (20024907, NextSeq 500/550 High Output Kit v2.5, 150 Cycles, San Diego, CA, USA). Both sequencing platforms were analyzed with defined criteria.

### 2.5. Bioinformatic Analysis

The raw sequencing results were aligned to the reference genome GRCh37/hg19 using a homemade pipeline (Humanomics v3.15, https://gitlab.uliege.be/bif-chu/humanomics, accessed on 2 June 2025 [20]) that utilizes published algorithms: BWA-MEM for mapping the reads, elPrep for filtering the reads and removing duplicates, HaplotypeCaller for detecting variants, and GenotypeGVCFs for producing variant-calling format (VCF) files. VCF files are used for the filtering and interpretation of variants. Humanomics allows identification of single-nucleotide polymorphisms (SNPs) and short insertions and deletions (indels, from 1 to 15 bp) located within exons or at the intron–exon boundary (~50 base pairs of flanking regions). The workflow does not call copy-number variants (CNVs), large deletions, mosaicism, or other structural variants; no CNV analysis was performed in this study due to insufficient positive controls for validation.

### 2.6. Sensitivity and Precision of Sequencing

To estimate sensitivity and precision, we used GIAB sample. This sample was sequenced and analyzed within the same workflow as the DNA extracted from the DBS. The generated VCFs were subsequently compared to publicly available data provided by the GA4GH consortium [21]. These reference data consist of a VCF encompassing all variants confidently identified in this sample, referred to as the GIAB gold standard, and a BED file containing all regions fully characterized for this sample. These regions, known as high-confidence regions (HCRs), cover approximately 85% of the genome (hg19) and serve as the standard reference regions. Real-time genomic (RTG) tools v3.12 (vcfeval) were used for VCF comparisons. Vcfeval from RTG tools serves as a tool for evaluating the accuracy of genetic variant calls. It compares two sets of variants: one extracted from the VCF being tested and another from a designated set file—the GIAB gold standard. The analysis was restricted to TOI, looking only at variants (i) present in the HCR and (ii) at positions covered with at least 30 reads. True positive (TP) represents variants found both in GIAB gold standard and in our VCF; false negative (FN) represents variants from GIAB gold standard that are missed in our VCF; and false positives (FPs) are variants present in our VCF but not in GIAB gold standard. Sensitivity is defined as the fraction of GIAB gold standard detected; hence, it is equal to TP/(TP + FN). Precision is defined as the fraction of variants from our VCF present in GIAB gold standard; hence, it is equal to TP/(TP + FP). These two metrics provide a quantitative measure of the reliability of variant calls.

### 2.7. Reproducibility of the Results

The pairwise comparison of samples from various groups was performed using RTG tools, using one VCF as test sample and the other as a “truth” set. We assessed concordance with the formula C/(A + B + C), where C corresponds to variants found in both VCF1 and VCF2 (hence classified as TP by Vcfeval), A corresponds to variants exclusively inferred in VCF1 (hence classified as FP by Vcfeval), and B corresponds to variants inferred only in VCF2 (hence classified as FN by Vcfeval).

The analysis was limited to the TOI and HCRs and filtered to retain only those with a minimum depth of 30 reads (DP ≥ 30).

### 2.8. Definition and Selection of Quality Metrics

To ensure the reliability and accuracy of our results, we check quality control (QC) parameters that span various stages of sample processing, from library preparation, sequencing to variant calling. Among the numerous QC parameters provided by the Picard tools suite [22], the ten have been chosen for longitudinal monitoring (Appendix A).

### 2.9. Threshold Setting

For every sample processed, QC values generated by Picard tools are systematically stored in a centralized database. While the complete set of metrics is extensive, only a curated subset is required to validate the quality of an analysis under routine conditions. A sample is flagged as potentially problematic when, for each retained metric, its inferred value is outside a defined quality threshold.

To define QC thresholds, 34 independent runs were considered for panel-v1, and 43 independent runs were considered runs for panel-v2. Duplicate and triplicate samples were excluded to avoid bias. For each metric, distribution plots were created to visualize the interquartile range (IQR) and determine cutoff values, with most parameters adopting a 1.5 × IQR threshold to identify potentially problematic samples.

For the longitudinal follow-up, we have adopted ChronQC as an integral tool for monitoring the quality of our data over time [23]. Leveraging ChronQC’s capabilities, we can efficiently track changes and trends in our datasets across multiple time points, ensuring the reliability and consistency of our longitudinal analyses. All quality metrics are monitored for each sequencing run. We have defined three very important metrics (VIPs): Q30_pct, TARGET_BASES_30X_pct, and SNP_REFERENCE_BIAS. In the case of failure of one of these VIPs, the whole workflow for that sample is repeated.

### 2.10. Variant Interpretation Pipeline

After the secondary analysis was completed and the sequencing quality was checked, the VCFs were used for variant interpretation of asymptomatic neonates, where no phenotypic data was available for the variant selection process. Using the Alissa Interpreter (Agilent), we have developed and validated the BabyDetect Variant Interpretation Pipeline. The pipeline includes (a) the decision tree that has criteria for filtering variants with no human phenotype ontology and (b) the information of genes in the panel [14].

In BabyDetect, we filter and report class 4 and 5 variants according to ACMG guidelines [24] from our knowledge-based database, as well as from ClinVar. Variants which are not present in ClinVar were not reported. Filtered P/LP variants were assessed manually using additional platforms of Franklin (https://franklin.genoox.com, accessed on 22 September 2025) and VarSome [25]. Variants with conflicting assertions of pathogenicity between these three platforms (ClinVar, Franklin and VarSome) were not considered for reporting.

## 3. Results

### 3.1. Workflow

The BabyDetect analysis was developed starting in 2020 and is now accredited under ISO-15189:2022 [26] certification and IVDR-labeled. Its analytical workflow is defined from sample punching at the laboratory facility up until clinical reporting (Figure 1). The mean laboratory processing time for workflow is from 8 to 10 days.

Samples for the study and for analytical validation were collected, recorded, and stored by trained personnel under conditions for clinical application. DBS cards were delivered to our dispatching unit and encoded with a unique identifier in the information management system.

Following sample reception, the analytical workflow begins with the punching of the DBS card. This step is followed by an initial QC evaluation; in the event of failure, due to insufficient material or poor sample integrity, the corrective action is initiated, including re-contacting the maternity ward or verifying the availability of an alternative card. If the sample passes QC, it proceeds to DNA extraction, which is likewise subjected to QC before advancing to library preparation. To optimize DNA yield, we initially incorporated an overnight lysis of DBS punches at 30 °C, followed by manual DNA extraction using the Qiagen column-based protocol. This approach was applied to validation samples and one-year screening samples. As the study progressed and the need for scalability and shorter TAT became evident, we validated and transitioned to automated extraction using the QIAsymphony platform. This automated method was subsequently adopted for the remaining study samples. Twist Bioscience technology is used for library preparation; the average size of prepared libraries with ligated adapters is 350 bp, which allows sequencing of fragments by both NovaSeq and NextSeq systems. At each stage, QC is systematically applied. QC failure at any point triggers a loopback to the extraction phase, ensuring that only high-quality input progresses through the workflow. This looped design not only safeguards the fidelity of downstream analyses but also minimizes the risk of reporting inaccurate findings. The final steps involve variant interpretation and clinical reporting. When a positive result is identified, the workflow is repeated from the punching of DBS until variant interpretation. This allows us to confirm the positive screening result as well as to check for inversion of the sample.

The workflow described in this section was used for analytical validation, whereas the optimizations for DNA extraction as well as a panel redesign are presented in a dedicated section below.

### 3.2. Variant Interpretation Pipeline

In BabyDetect, for conditions with autosomal-recessive inheritance, we report homozygote and compound heterozygote P and LP variants. Carriers of those variants are not reported in the scope of our project. In case of autosomal-dominant diseases, the presence of one P or LP variant is reported. For X-linked diseases, hemizygous identification of P or LP variants in males and homozygous or possible compound heterozygous identification of P or LP variants in females are reported [14]. The variants of uncertain significance (VUSs) are not reported.

Our decision tree incorporates different filtering criteria, including quality, allelic frequency, mutation type, and operates to decipher variants of clinical interest (Figure 2). Briefly, after secondary analysis and sequencing quality verifications, VCFs are uploaded in the interpretation pipeline. Variants are first filtered based on our gene panel. Then, variants are filtered by read depth and variant confidence by depth. Filtering based on variant allele and population frequency is then applied. Variants with population frequency <1% or any variants in frequent genes are checked in our managed variant list (MVL). Variants that match P and LP classification from MVL are selected for filtering based on X-linked genes in the panel and subsequently filtered by their zygosity. Furthermore, variants are filtered on published P and LP ClinVar variants, and matching variants are again filtered based on X-linked genes and zygosity. All filtered homozygote, compound heterozygote, and hemizygote variants are manually reviewed.

The filtering pipeline was designed with a semi-automatic approach; when there are no variants of interest to review, the sample is automatically closed, whereas samples with P and/or LP variants are flagged for manual review. Variants are reviewed independently by three scientists and decisions on reporting are taken. Variants not registered in ClinVar or in our MVL are not reported. Variants with conflicting interpretations in ClinVar, Franklin, VarSome databases and relevant publications are checked. Consensus on reporting is taken if supporting evidence is found in three databases.

### 3.3. Validation Samples

Initial validation was performed on a GIAB reference (HG002-NA24385) and on eight positive samples previously identified by an alternative method. An additional eight newborns’ negative samples and four adults’ negative samples were used for comparative analysis of different parameters of the workflow. These adult whole blood samples with no reported conditions were also spotted as DBS and included in the validation process, to assess concordance between our DBS workflow and classical whole blood workflow.

Sample selection, included in the validation plate, is described in Section 2 Material and Methods. For initial validation (panel-v1), we used the first eight positive samples (NBPOS-1 to NBPOS-8) indicated in Appendix A; these samples were selected by their availability at the conventional NBS laboratory, which were positive based on biochemical marker and/or identified P variants at diagnostic setting. A total of 88 samples, included in the validation plates (Appendix A), were sequenced twice in independent sequencing runs on NovaSeq 6000 and 48 samples were sequenced on NextSeq 500/550.

After validation, 2600 newborn samples included in the project were processed with the validated protocol in 34 independent batches (96 or 48 samples/batch). The protocol with a redesigned panel (v2) was revalidated on a set of eight additional positive samples (Appendix A) and the same GIAB. These eight samples were sequenced on two independent sequencing runs. This second protocol was then used for the remainder of the study (3319 samples in 43 independent batches).

### 3.4. Performance of the Analysis

#### 3.4.1. Sensitivity, Precision

Our results indicate that the initial validated workflow exhibits high sensitivity for SNP detection, with values exceeding 97% and a precision of 94% (panel-v1); whereas for the detection of indels, sensitivity is 81% and precision is estimated at 59% (Figure 3).

Our findings underscore the method’s high sensitivity and precision for SNP detection, together with its reliable sensitivity but moderate precision for detecting small indels. Our obtained results were in accordance with the number of accredited NGS analyses of the diagnostic unit at the university hospital of Liege (CHU Liege, oral communications at laboratory meetings and validation protocols), both for panel-based sequencing and whole-exome sequencing (WES).

The redesign of our target panel (v2) and its improved performance are described in the section “Optimizations” below.

The BabyDetect Variant Interpretation Pipeline was validated on positive and negative samples (*n* = 216 sample). In the initial analytical validation, eight positive samples were analyzed in triplicate in two NovaSeq runs (48 samples) and, in duplicates, in a NextSeq run (16 samples). All disease-causing variants, identified by alternative methods, were correctly selected by our variant decision tree which leads to a 100% sensitivity for P variants of our workflow. Negative samples included in the validation plates were also interpreted with the pipeline and no P variants were retained after interpretation (Novaseq runs *n* = 112 samples, Nextseq run *n* = 24 samples) as expected.

The eight positive samples used in revalidation (panel-v2) of the workflow were sequenced in two runs (*n* = 16 samples) and interpreted with the interpretation pipeline; the same P variants were identified in both runs with 100% sensitivity.

#### 3.4.2. Intra-Run and Inter-Run Concordance

We have used the concordance as an estimation for the similarity between two replicates of a sample within the workflow.

For GIAB, the estimated intra-run mean concordance between replicates of the same batch was 93%. The concordance analysis was also performed for positive controls and the control extracted from whole blood and DBS (intra-run mean concordance—90%).

Inter-run concordance between the two replicates passed in two independent batches was used to estimate their similarity. Our results show an inter-run concordance close to 90% for SNPs for GIAB and positive controls. The majority of discordant variants (92%) were FP with low allele frequencies, and are not relevant for clinical interpretation, as they are systematically excluded by our variant filtering pipeline.

Overall, these results indicate a high reproducibility of our workflow.

### 3.5. Quality Control

To ensure the reliability and accuracy of our sequencing data, we established a set of QC thresholds. Samples that do not meet these thresholds are flagged for re-analysis, ensuring data integrity and consistency. The quality metric thresholds for both panel-v1 and panel-v2 used in routine are presented in Appendix A.

#### 3.5.1. Evaluating Sequencing Quality

To ensure the reliability of the sequencing data, we routinely monitor key quality metrics that reflect both yield and accuracy. We used core indicators PF_BASES and Q30_pct.

A minimum threshold of 85% (panel-v1) is applied for Q30_pct as recommended by the instrument manufacturer. Based on revalidated panel-v2 and the rest of the sequenced runs, the minimum threshold was raised to 90% to further increase the stringency of our QC (Appendix A). All samples in our cohort exceeded these values, confirming the high quality of our sequencing runs.

For PF_BASES, we established a lower limit based on inter-sample variability observed across multiple validation runs. Using the IQR as a robust measure of spread, we defined the threshold as 1.5 × IQR below the cohort median—an approach used to flag statistical outliers.

Samples below this limit are flagged for review. If a sample fails Q30_pct it is excluded from further analysis and is re-prepared from the newly punched DBS. This strategy helps to ensure that only high-quality data is retained for downstream interpretation.

#### 3.5.2. Evaluating Target Selection Quality

To monitor library preparation, we used TARGET_BASES_30X_pct and MEAN_TARGET_COVERAGE metrics.

Initially, the TARGET_BASES_30X_pct metric had a lower threshold set at 93%. Samples falling below this threshold exhibit insufficient coverage and are systematically reviewed and the flow is repeated for that sample (Figure 1). When MEAN_TARGET_COVERAGE also falls outside the acceptable range, this strongly indicates inadequate sequencing depth, warranting review. The MEAN_TARGET_COVERAGE threshold is arbitrarily set at 100×, ensuring robust depth across the targeted regions.

To evaluate the capture efficiency in library preparation, we follow the QC metrics of SELECTED_BASES_pct. For initial panel -v1, the minimum threshold of this parameter was set to 40% (Appendix A). The redesign of the panel (v2) allowed us to improve the on-target capture (min. threshold 78%), and the details are described in the dedicated section below.

#### 3.5.3. Evaluating Inversions and Contaminations

In our routine workflow, to control plate or sample inversion during manual preparation, we have introduced two risk mitigations:

Every 96-well plate includes in position H12 a control, containing P variants in two genes (*SERPINA1*: c.1096G>A and *ALDOB*: c.448G>C). This sample serves as the “internal control sample”.

Each positive case is completely reanalyzed from the original DBS to confirm the initial sequencing finding.

Additionally, we use the SNP_REFERENCE_BIAS to detect potential contamination. The upper threshold of the SNP_REFERENCE_BIAS is fixed at 0.56, corresponding to 1.5 × IQR above the median. Values exceeding the threshold suggest contamination and are reviewed and re-prepared.

The “internal control sample” is monitored after each variant interpretation of the batch to ensure that P variants are identified in every sequencing run with sufficient read and allelic depth, call and mapping quality, and true genotype. For each monitored parameter, the Levey–Jennings chart is generated, and the values are plotted after each batch interpretation. This monitoring allows us to confidently state that for that sample both P variants are being identified with high confidence and uniformity (Appendix A).

#### 3.5.4. Longitudinal Monitoring

We use ChronQC (Figure 4) to automate the monitoring and evaluation of the sequencing metrics over time, ensuring that the data consistently adheres to predefined quality standards throughout the study. We monitor all the metrics outlined in the ChronQC report. The samples failing the defined thresholds for the VIPs are being repeated from the start of the laboratory workflow.

#### 3.5.5. Decision Criteria for Sample Quality

Across the first year of the study using panel-v1 (2600 samples), 95.6% (*n* = 2486) of samples passed all QC criteria without requiring the repeating of the workflow. The remaining 4.4% (*n* = 114) failed at least one metric: 103 samples failed one metric, 10 samples failed two metrics, and 1 sample failed three. The most frequent failure was SELECTED_BASES_pct (*n* = 82). By systematically integrating these thresholds into our workflow, we enhanced both sequencing efficiency and downstream data accuracy.

### 3.6. Robustness to Workflow Variations

We have assessed whether variations in the standard conditions of the protocol for library preparation have any influence on results quality.

#### 3.6.1. Initial DNA Quantity

Initial DNA quantity can vary due to DNA extraction and manual manipulation. To evaluate the robustness of our workflow to such DNA quantity variations, we estimated sensitivity and precision for the detection of known GIAB variants using DNA quantities of 40 ng and 50 ng for library preparation as recommended by Twist Bioscience. Our results indicate that both DNA quantities lead to a similar sensitivity for SNP detection (97.15%-40 ng and 97.21%-50 ng) and precision (94.08%-40 ng and 93.67%-50 ng). Sensitivity and precision for indels are also similar between both DNA quantities (sensitivity at 81.58%-40 ng and 81.49%-50 ng, precision at 59.54%-40 ng and 58.92%-50 ng) (Figure 3).

Furthermore, to facilitate the manual workflow we did not quantify the amount of DNA to be taken for library preparation but took 40 uL of extracted DNA for fragmentation. Of note, extracted DNA concentration varied from 50 to 400 ng. We have tested the robustness of the analysis by varying the DNA input of 40 ng vs. 50 ng and 40 µL of extract vs. 40 ng. Results show that the concordance observed here does not differ from the inter-run concordance (90%). The amount of DNA used for library preparation has little impact on concordance. This demonstrates consistency and shows that our method is robust to variations in initial DNA quantity.

#### 3.6.2. Initial Material Variation: Whole Blood or Dried Blood Spot

Sensitivity and precision were evaluated on GIAB but our routine flow for neonatal screening uses DNA extracted from DBS [27]. Therefore, we assessed concordance between DNA extracted from whole blood and the same DNA extracted from DBS. Results show that the concordance observed here does not differ from the concordance observed in the inter-run concordance data (90%) and that results obtained from DBS are no different from those of whole blood.

#### 3.6.3. Sequencing Instrument

Depending on the number of samples to screen, we have used both NovaSeq (96 samples) and NextSeq (48 samples) sequencers. To evaluate whether sequencing instruments influence the quality of obtained results, we calculated concordance, precision, and sensitivity for the GIAB between NextSeq and NovaSeq, for DNA quantities of 40 and 50 ng. The sequencing runs on NovaSeq are referenced as NovaSeq_1 and NovaSeq_ 2, the NextSeq run is referenced as NextSeq_1.

The results show that the concordance between NovaSeq and NextSeq fluctuates approximately between 86 and 88% for the control samples, whereas the concordance for the GIAB sample was lower, ranging between 83% and 85%, with no difference between DNA input 40 and 50 ng.

### 3.7. Optimizations

#### 3.7.1. DNA Extraction Automation

To ensure scalability and to improve TAT, the automated QIAsymphony DNA extraction was validated and implemented in our workflow. It has since replaced the manual extraction of our routine workflow, and 3319 samples have been screened.

For both extractions, the amounts of isolated DNA fluctuated between 50 and 400 ng, with a DNA size greater than 20 kb with the manual column-based method (Appendix A-b) and greater than 40 kb with the beads-based QIAsymphony (Appendix A-a) extractions. Concentrations obtained for selected DNA-plates are available in Appendix A. ANOVA test was conducted, and it showed a *p*-value = 0.188 (*p* < 0.05), indicating that there is no significant difference between DNA concentrations extracted by both methods. Additionally, after running extracted samples with the entire analytical flow, pairwise concordance (per sample) was calculated between samples extracted manually and with QIAsymphony. Samples were sequenced in different runs and the results are presented in Appendix A. The mean concordance between analyzed samples is 91.6. The comparisons of both methods allowed us to integrate, with confidence, the QIAsymphony workstation into our further flow.

#### 3.7.2. Target of Interest: Improving Performance and Clinical Impact

After the first year of this study, we re-evaluated the gene list of our targeted panel and removed and added some genes (Appendix A). The re-evaluation of gene list meant a redesign of the panel. This opportunity of redesigning has allowed us to improve our general performance by focusing on the two following aspects. First, we removed some genes because they could not meet our quality control criteria due to technical limitations (homologous regions, large rearrangements). Second, we reduced the number of targeted regions (namely by removing large intronic regions) to reduce the number of off-target reads. The initially validated panel of BabyDetect contained 359 genes and the redesigned panel has 405 genes. Precisely, 15 genes were removed from the initial panel and 61 were added to the redesigned panel.

Beyond its broader screening scope, the redesigned panel demonstrated improved technical performance. A key indicator of this improvement is the significant reduction in off-target sequencing: the average percentage of off-target reads decreased from 52.7% with panel-v1 to 18.5% with panel-v2, representing a ~30% relative reduction (Figure 5). This improvement in on-target efficiency is critical (SELECTED_BASES_pct), as off-target reads can obscure variant interpretation and reduce overall data quality. By minimizing off-target capture, panel-v2 not only improves hybridization specificity but also enhances the reliability of downstream variant calling.

In line with these technical optimizations, we also observed a marked improvement in variant detection performance with panel-v2. While the initial workflow using panel-v1 already demonstrated high sensitivity for SNP detection, >97%, and a precision of 94%, the redesigned workflow with panel-v2 yielded even better results (Figure 3). Across all replicates, SNP sensitivity reached 99%, with precision ranging from 95% to 97%, reflecting more accurate identification of true positive variants. Although indel calling remained more variable, slight improvements were also observed in panel-v2, with sensitivity ranging from 79% to 83% and precision ranging from 56% to 64%.

These improvements at the panel design level complement the performance gains observed during the revalidation of our workflow and contribute to a more robust and clinically impactful screening tool.

It is important to indicate that, to further analyze the identified low precision of indels, we have investigated false positive indels in a GIAB sample and provided results in the Appendix A. Specifically, we identified and investigated 51 indel variants from the HG002-NA24385 reference sample captured with panel-v2. Most variants were located at exon–intron boundaries; a few were within the same gene and only a few were annotated in databases. The majority of these variants had low allele frequencies and/or were classified as benign with no clinical relevance for interpretation and thus were systematically excluded by our variant filtering tree. For this reason, we accept the relatively low precision values for indels.

## 4. Discussion

In the BabyDetect project, we currently employ a tNGS panel encompassing 405 genes associated with 165 diseases (panel-v2). Our approach delivers high sensitivity and consistent detection of SNPs across all replicates, with robust precision metrics. As expected, detection of small insertions and deletions exhibits greater variability, particularly in regions of low sequence complexity, capture inefficiency, or homopolymers—challenges intrinsic to enrichment-based capture techniques. Indeed, indel calling remains technically more complex than SNP calling, as enrichment protocols introduce elevated indel error rates, especially in A/T-rich homopolymer stretches.

A key limitation of the present manuscript is the absence of copy-number and structural variant analysis. We intentionally restricted scope to SNVs and short indels to maintain screening-grade positive predictive value (PPV), batch-to-batch stability, and turnaround time, given the lack of sufficient positive controls for pipeline validation. Indeed, SNVs represent the predominant cause of genetic disease, accounting for an estimated 85% of known pathogenic mutations [28,29]. In comparison, CNVs and other SVs contribute to approximately 10–15% of genetic disorders [30]. CNVs, defined as gains or losses of DNA segments, can markedly alter gene dosage and function. Although CNVs span 12–16% of the human genome, only a subset are considered rare and clinically relevant [31]. These rare CNVs account for approximately 10% of SVs implicated in rare diseases [32]. However, not all rare pediatric disorders are equally enriched for pathogenic CNVs. Syndromic neurodevelopmental disorders (e.g., intellectual disability/global developmental delay with congenital anomalies or epilepsy) often exhibit high CNV yields (~20–30%), though SNVs and small indels contribute majorly to disease [33,34]. Moreover, certain disorders, such as Duchenne muscular dystrophy or Angelman syndrome, are CNV-driven by definition, with CNVs representing the primary pathogenic mechanism (~70–80%). As such CNV-driven conditions were excluded from our screening gene list, based on our eligibility criteria, we have 17% of genes (68 out of 405 genes) that contain reported CNVs and other SVs, and in most of them the estimated contribution of pathogenic CNVs/SVs is approximately 2–5%.

As a prerequisite to routine deployment, we will conduct a staged validation using confirmed CNV-positive samples that reflect the diversity of events likely to occur in our assay: (i) single-exon versus multi-exon deletions/duplications; (ii) small versus large events in terms of the number of captured targets; (iii) loci with high homology or pseudogene interference; (iv) GC-rich and GC-poor segments; and (v) exons at capture boundaries. Acceptance criteria will include per-gene/region sensitivity estimates, PPV compatible with population screening, a stable per-sample call burden, and inter-run reproducibility. Upon meeting these criteria, CNV analysis will be activated in routine, with MLPA or qPCR confirmation required for any putative screen-positive sample prior to clinical reporting.

To mitigate the current scarcity of confirmed positives, we will also run a prospective, off-line “shadow” CNV evaluation across all incoming samples under a research protocol separate from routine screening and reporting. This will not affect turnaround time or clinical outputs. Candidates will be restricted to high-confidence patterns (e.g., multi-exon events in well-covered genes, consistent signal across QC checks with acceptable sample-to-reference correlation and a non-inflated per-sample CNV burden). Any candidate meeting the predefined triage criteria will undergo orthogonal testing on retained material (MLPA/qPCR). Confirmed events will be used to refine thresholds, estimate gene-level sensitivity/PPV, and lock a sufficiently large panel-of-normals for robust normalization. Once screening-grade performance is demonstrated, CNV analysis will transition from shadow evaluation to routine initially on a CNV-ready subset of genes.

Improving capture efficiency can enhance indel sensitivity in tNGS and WES. Yet, whole-genome sequencing (WGS) offers a clear technical advantage by enabling more uniform coverage of the genome, with estimates suggesting that 60× is needed to recover 95% of indels [35]. WGS further extends these capabilities by including both coding and noncoding regions. This comprehensive view enables the detection of structural variants, deep intronic and regulatory mutations, repeat expansions, and CNVs with higher sensitivity and resolution than targeted- or exome-based approaches. Such breadth is particularly advantageous to increase diagnostic yield in neonatal screening programs, especially in disorders characterized by heterogeneous genetic architectures [36,37].

The transition from our initial panel (v1) to panel v2 was driven by the need to increase diagnostic yield by including newly validated disease genes and optimizing capture performance. Each modification of the panel, whether the inclusion of additional regions or the removal of poorly performing ones, requires extensive revalidation, which is resource-intensive and time-consuming. These repeated validation cycles can create bottlenecks in implementation and, in some cases, lead to sample backlogs. Retrospectively, the use of WES or WGS could have accelerated this development phase by enabling broader exploratory variant detection, reducing the need for iterative panel redesigns. WES enables the analysis of nearly all protein-coding regions of the genome, which harbor the majority of known disease-causing variants [38]. This allows for the detection of a wider spectrum of pathogenic single-nucleotide variants (SNVs), small indels, and, with appropriate bioinformatic tools, some larger CNVs [39]. Nevertheless, the high per-sample cost and limited throughput of WES and WGS remain significant barriers to its use in population-scale screening programs, while also considering the limitations associated with sequencing a large number of samples on a single flow cell [40,41,42]. Additionally, the costs of extrapolating WES or WGS on the national scale underscores the current economic impracticality of these broader approaches for routine screening. Furthermore, beyond cost, these genome-wide approaches also introduce significant challenges. First, they generate a vast amount of data, increasing the burden of computational analysis, storage, and interpretation, especially in a time-sensitive context such as neonatal care [36]. Second, they yield a higher proportion of VUS, which complicates clinical decision-making and can lead to ethical dilemmas regarding disclosure and follow-up [24]. Moreover, the incidental identification of secondary findings, which are not related to the primary indication for testing, raises additional ethical and logistical concerns regarding consent, counseling, and long-term follow-up [43]. In contrast, targeted panels allow for streamlined analysis, reduced turnaround time, and alignment with established clinical actionability frameworks, making them more immediately implementable in public health settings.

Therefore, while WES and WGS hold promises for future implementation, especially in cases where a broader genetic investigation is warranted, current evidence supports the continued use of targeted panels as a pragmatic and scalable solution for population-wide neonatal screening. Strategic integration of CNV detection into panel-based methods, through algorithmic enhancements or supplemental assays, may further improve diagnostic performance without compromising operational feasibility.

It is important to note that although second-generation short-read sequencing is efficient and cost-effective, it lacks capacity to phase pathogenic variants without parental samples to assess segregation (cis or trans). The need to request parental samples can prolong the diagnostic process and cause anxiety for families while results are pending. Third-generation long-read sequencing platforms offer a solution to this challenge. A major advantage of long-read sequencing is its ability to provide phasing information directly from a single individual, linking variants located on the same DNA strand. This approach removes the requirement for parental DNA to determine variant inheritance in recessive conditions, thereby reducing diagnostic delays, alleviating parental anxiety, and streamlining laboratory workflows. Moreover, long-read sequencing enables more accurate detection of CNVs and other structural variants.

Variant filtering and interpretation remain challenging for genomic NBS, especially in the case of neonatal screening in a considerably healthy population where no phenotypic data is accounted. Our developed filtering strategy and conservative approach, which reported only known pathogenic and likely pathogenic variants, enabled us to identify 1.8% of screened cases as positives, with 0.8% not identified by conventional screening [14]. Filtering of pathogenic and likely pathogenic variants based on only known databases (ClinVar, Franklin, VarSome) and our own MVL may increase the number of clinical false negatives (1 false negative out of 71 positives [14] and 2 false negatives out of 114 positives—BabyDetect unpublished data). Alternatively, reporting VUS still remains challenging and is a subject of debate. In the context of newborn screening, overloading the variant interpretation pipeline with VUS variants and reviewing them manually can cause increase in screening TAT, whereas reporting of VUS can cause anxiety among the screened population. Our developed filtering pipeline allows us to keep VUS in a separate location in the database while not reviewing and not reporting them. The VUS datasets could be used for future re-classification in case of available proof of pathogenicity and/or new scientific projects.

To define an appropriate balance between false negatives and false positives in BabyDetect, we assessed the trade-off between variant review workload and TAT, especially critical for the early onset of treatable conditions. A higher number of variants flagged for manual review, especially VUS, would drastically increase the TAT, potentially delaying clinical intervention during the neonatal window where treatment timing is essential. Conversely, overly stringent filtering reduces the number of variants to review but increases the risk of false negatives. We therefore adopted a filtering strategy that minimizes manual burden without compromising sensitivity for known actionable conditions. This compromise was refined through retrospective analysis of early cohorts and benchmarking against conventional screening results, allowing us to calibrate filters to optimize both diagnostic yield and operational feasibility. Manual review is strictly limited to variants meeting defined pathogenicity criteria, and VUSs are stored for future re-analysis, preserving both clinical rigor and workflow efficiency.

The main challenge for variant interpretation associated with asymptomatic newborns undergoing newborn screening is the absence of symptoms that can be predictive for any suspected disease. In BabyDetect, we implemented a structured variant filtering tree, not considering the human phenotype ontology, which provides a standardized vocabulary of phenotypic abnormalities encountered in human disease. Our standardized framework integrates automated filtering, pathogenicity scoring based on ACMG/AMP guidelines, and manual curation steps. Each node of the decision tree was carefully designed to incorporate both objective evidence (e.g., variant frequency, gene–disease association, and expert-driven judgment). This architecture ensures reproducibility while preserving clinical oversight, especially for conflicting variants near the classification boundary between VUS and the likely pathogenic.

There is another challenge associated with variant interpretation pipelines, such as the use of artificial intelligence (AI)-based tools. Not underestimating the benefits associated with the implementation and use of AI in variant filtering (exp. Franklin platform, https://franklin.genoox.com, accessed on 22 September 2025), it is important to have sufficient knowledge to correctly consider and analyze, evidence-based, the classification provided by AI tools for each filtered variant. In addition, as AI tools become more widely adopted in genomics, the field of explainable AI (XAI) has emerged as essential, particularly in clinical contexts where understanding the rationale behind a prediction is as important as the prediction itself. In variant interpretation, XAI approaches can enhance transparency by highlighting which features contributed most to a classification—such as functional annotations, conservation scores, or population data—thus allowing human experts to validate or challenge the output. This is especially relevant in NBS, where decisions must be timely, ethically sound, and clinically actionable. Recent efforts have underscored the importance of interpretable models in healthcare applications [44] and in variant classification workflows [45].

For laboratories aiming to implement newborn genomic screening, the integration of explainable AI into decision-support systems offers a promising path forward, but it must be coupled with domain expertise, rigorous validation, and transparent governance structures. Defining thresholds for actionability, clarifying roles of AI outputs, and preserving the option for expert override are critical safeguards to ensure that automation enhances, rather than replaces, clinical judgment.

In the context of population-level screening, where clinical sensitivity and specificity are paramount, the bioinformatics pipeline must not only be accurate but also stable, auditable, and reproducible across time and environments [46]. Our current bioinformatics pipeline is under strict revision control with in-silico revalidation of every major release, and execution is run on a dedicated high-performance cluster with containerized software tools under revision control as well. Our pipeline is implemented using a series of Bash scripts. While this legacy approach has been carefully maintained and regularly updated, it lacks features such as integrated workflow management, native scalability, and environment portability. As part of our continuous improvement strategy, and in line with our practices for other genomic workflows, we are planning to migrate this pipeline to a workflow manager framework. This transition will improve modularity, reproducibility, containerization, and long-term maintainability, particularly critical in the context of clinical deployment at scale. We thus advocate for a modular, containerized architecture built with workflow managers such as Nextflow [47] or Snakemake [48], which allow explicit version control of all steps, tools, and reference files. Containerization (e.g., via Singularity [49,50] or Docker [51]) ensures computational reproducibility by isolating the runtime environment and avoiding hidden system-level dependencies. Each step in the pipeline—from base calling and demultiplexing to alignment, variant calling, annotation, and report generation—should be fully logged, traceable, and testable.

Our experience in deploying in-house pipelines under ISO 15189 accreditation has underscored the importance of rigorous development practices and traceability. All tools and custom scripts are versioned using Git, and changes are tracked through pull requests and structured code review. Every production release is associated with a changelog, and all parameter settings and reference files used during clinical runs are archived in immutable configurations. The use of containerization, primarily via Singularity, which is compatible with HPC environments, has proven essential for ensuring identical results across environments, which is a prerequisite for re-accreditation and external audits.

To support continuous integration and deployment, we have implemented automated test suites covering both unit-level validation of individual pipeline modules and integration-level checks using synthetic and well-characterized reference datasets. This approach allows us to detect regressions early, validate infrastructure updates, and minimize downtime during production transitions. Promoting pipeline changes from development to clinical production is governed by a formal approval workflow, which includes performance benchmarking, documentation updates, and review by a multidisciplinary team.

Importantly, we propose:Immutable configuration tracking, where every analysis run is associated with a fixed pipeline version, tool versions, and parameters.Automated unit and integration testing for all pipeline components, ensuring that updates or infrastructure changes do not introduce regressions.Reference dataset benchmarking to regularly evaluate the pipeline against synthetic or known truth sets (e.g., Genome in a Bottle, synthetic mixtures), thereby safeguarding analytical performance.Clear separation between development and production environments, with a formal promotion workflow when pipeline changes are validated and ready for deployment.Data provenance mechanisms (e.g., checksums, sample lineage tracking) to ensure that outputs can be backtracked to raw data and initial parameters.Furthermore, harmonization with external clinical guidelines should be embedded where applicable, particularly at the variant filtration and prioritization stages.

## 5. Conclusions

The BabyDetect project demonstrates that tNGS-based newborn screening is a reliable and scalable approach to detect treatable rare diseases not covered by conventional methods. Our validated workflow achieves high sensitivity and precision, with robust QC ensuring reproducibility across thousands of samples. Automation and panel optimization have improved efficiency and coverage. By focusing on clinically actionable variants, we balance diagnostic yield with manageable interpretation workload. While CNV/SV analyses were beyond the scope of the present work, we have outlined a staged plan for their integration in future studies: accrue positive controls for validation, establish a robust normalization cohort, and activate CNV calling with mandatory orthogonal confirmation once screening-grade acceptance criteria are satisfied.

## Figures and Tables

**Figure 1 IJNS-11-00091-f001:**
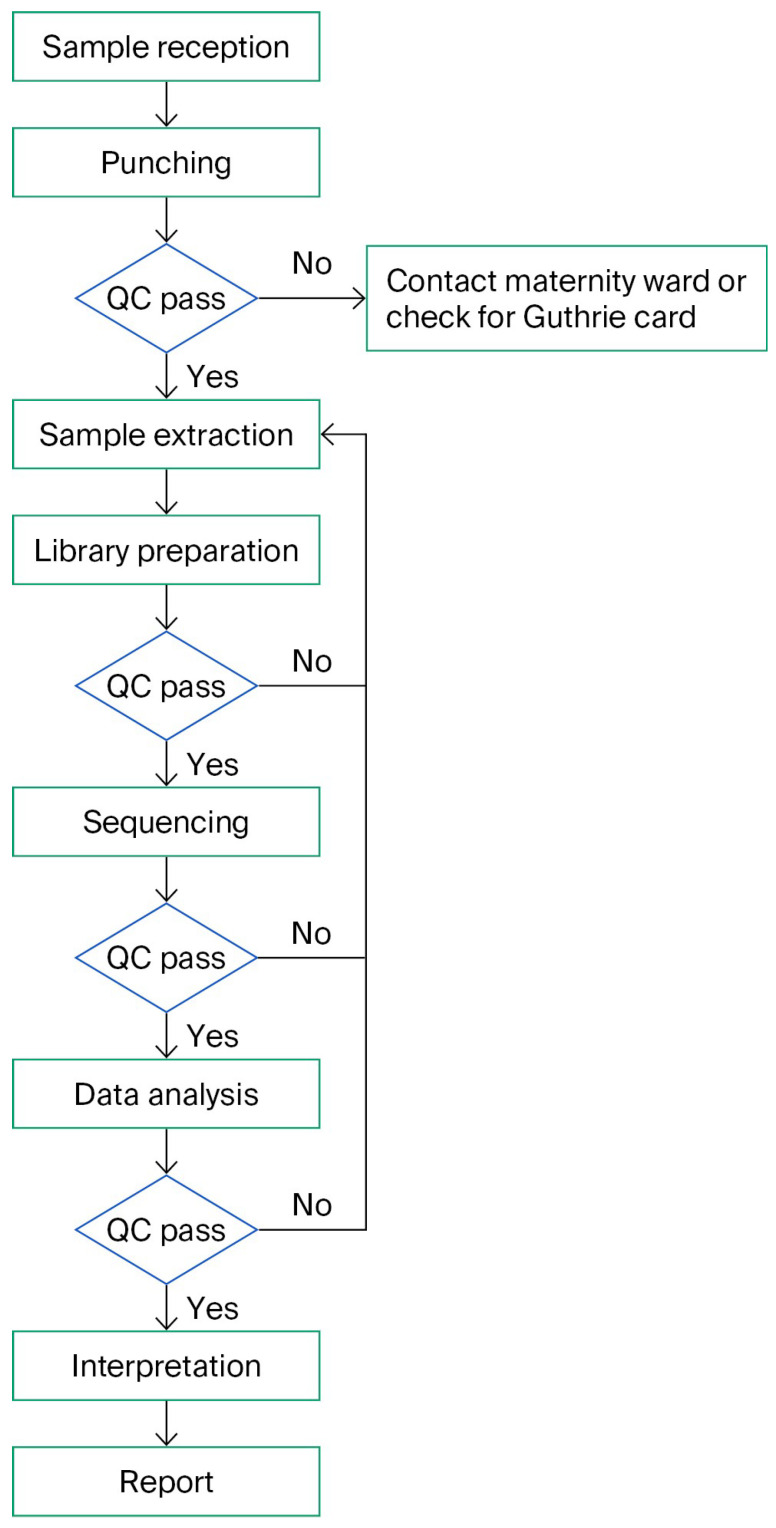
Laboratory operational workflow of BabyDetect analysis. Green boxes represent working steps in the flow from top to bottom. Blue diamonds represent quality control steps which lead to decision points on how to proceed.

**Figure 2 IJNS-11-00091-f002:**
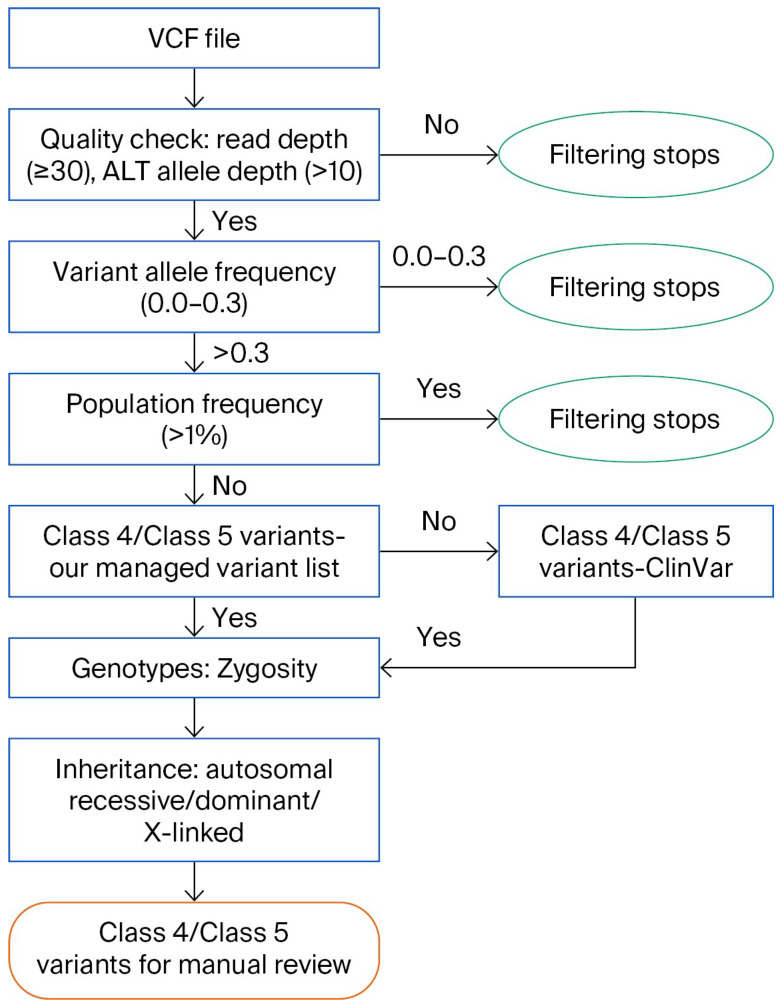
Decision tree for variant interpretation. Blue-marked boxes represent the filtering criteria, ovals indicate that filtering is not continued, and orange-marked box indicates the end of interpretation and manual review step.

**Figure 3 IJNS-11-00091-f003:**
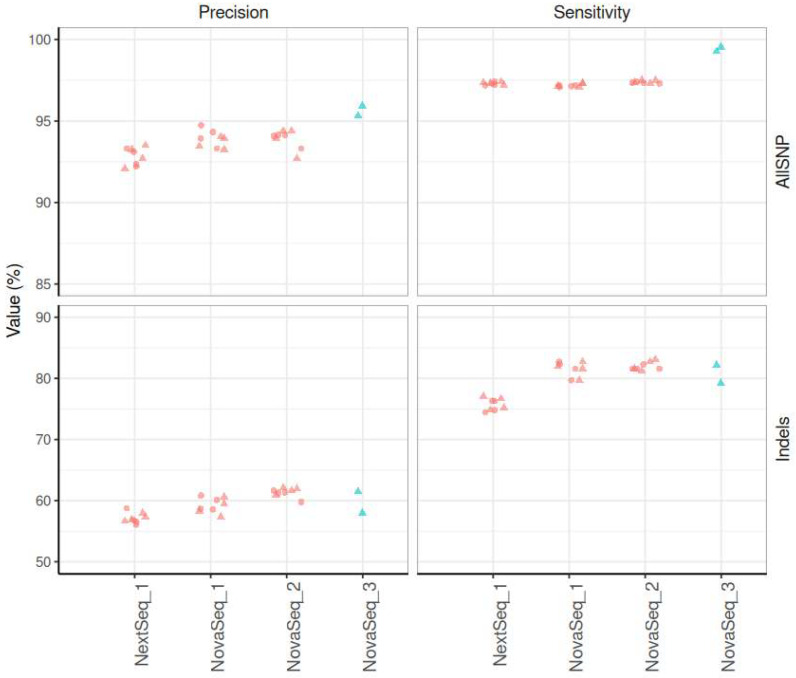
Performance metrics for SNPs and indels across sequencing panels and DNA inputs. Sensitivity and precision are shown for SNPs and indels across different sequencing panels and DNA input quantities. Upper part of the figure shows the Precision (left) and Sensitivity (right) metrics for SNPs. Bottom part of the figure shows the Precision (left) and Sensitivity (right) metrics for indels. Colors indicate the sequencing panel (red = Panel-v1, blue = Panel-v2), and shapes represent the input DNA quantity in ng (the circle shape corresponds to 40 ng of DNA, and the triangle shape corresponds to 50 ng of DNA). Facets separate metrics (columns) and variant types (rows). *Y*-axis scales are adapted per variant type.

**Figure 4 IJNS-11-00091-f004:**
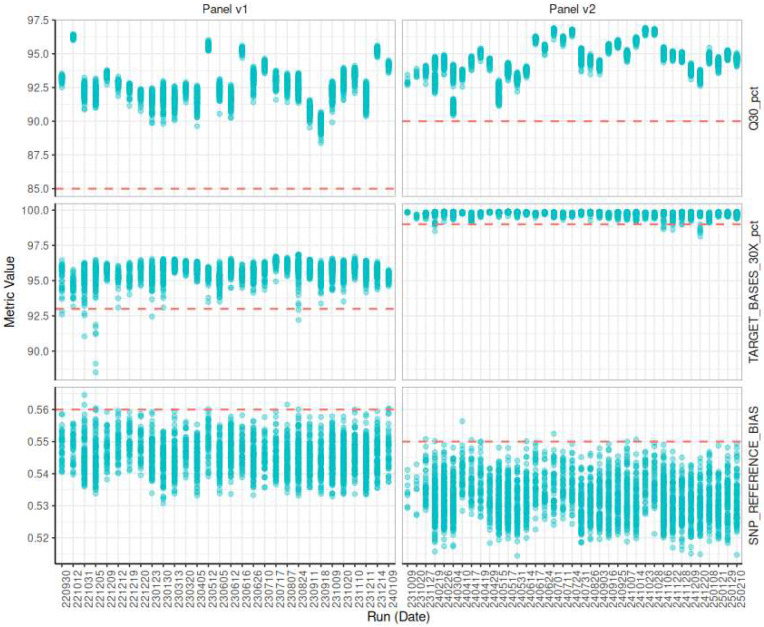
Sequencing quality metrics across runs for Panel v1 and Panel v2. Quality control metrics are shown for sequencing runs performed using two capture panels: Panel v1 (left) and Panel v2 (right). Each subplot displays one of three VIPs (top to bottom): the proportion of bases with a quality score ≥30 (Q30_pct), the percentage of target bases covered at ≥30× (TARGET_BASES_30X_pct), and the SNP reference bias (SNP_REFERENCE_BIAS). Each dot (blue) corresponds to an individual sample within a run, while horizontal dashed lines (pink) indicate the predefined quality threshold for each metric and panel. On the *x*-axis, runs are labeled by their run date; on the *y*-axis, metric values are shown.

**Figure 5 IJNS-11-00091-f005:**
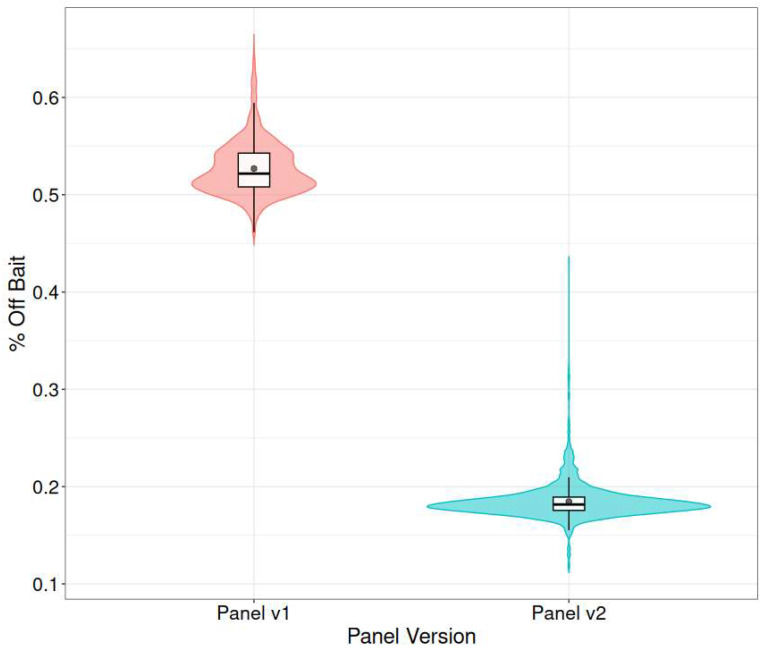
Distribution of off-target sequencing reads (% off bait) for exome capture panels v1 and v2. Violin plots represent the full distribution of % off bait across samples for each panel. Box plots indicate the median and interquartile range, and black dots denote the mean. Panel v2 (blue, right) shows consistently lower off-target rates compared to panel v1 (red, left), reflecting improved target enrichment performance.

## Data Availability

In accordance with the informed consent agreements, the raw sequencing data can be stored for each patient for 10 years. Metadata files are retained with no time limit. The raw sequencing data and metadata files generated in the study cannot be made publicly available because of ethical and data protection constraints. Deidentified data that support the results reported in this article will be made available to suitably qualified researchers through any requests that comply with ethical standards to the corresponding author (K.H., khovhannesyan@chuliege.be). Data must be requested between 1 and 12 months after the paper has been published, and the proposed use of the data must be approved by an independent review committee identified for this purpose by mutual agreement. Requests will be forwarded by the corresponding author to the identified ethics review committee. Upon acceptance by that committee, deidentified data will be provided by the corresponding author to the applicants through a secure web platform within 2 months. The minimum dataset required to run our code and reproduce results is available via Zenodo at https://doi.org/10.5281/zenodo.13935241 (accessed on 22 September 2025) [20].

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
