# Peer review of "Analytical Validation of a Genomic Newborn Screening Workflow"

_2409-515X, 2025, doi:10.3390/ijns11040091_

Round 1
Reviewer 1 Report
Comments and Suggestions for Authors
This manuscript presents the analytical validation of a targeted next-generation sequencing (tNGS)-based newborn screening workflow (BabyDetect), including data from over 5,900 (although 6,500 is also mentioned) samples and describing quality control (QC), variant interpretation, and workflow reproducibility. The scope and work is commendable.
Recommendation: Major Revision
The manuscript is technically well executed and provides a thorough analytical validation of a genomic newborn screening workflow. However, several significant methodological gaps—including lack of CNV/SV detection, limited indel precision, unphased variant calls, and overconservative interpretation filters—warrant revision before the study can be considered for publication. Detailed quantification of diagnostic limitations, transparent data sharing, and clearer methodological justifications will substantially improve the manuscript. The authors really need to focus on what should stay in the main body of the manuscript and what should be in supplement. This reorganization will help as currently it looks disjointed and difficult to follow. Below is a bit on the CNV area and these types of issues are seen throughout the manuscript in other areas as listed above briefly. As authors and reviewers we have a responsibility that we do our best to ensure the readers learn from the work and the experience of this team. Today many do not yet understand the NGS space well and they still think in diagnostic mode. The nuances of screening must come through. Having said that here is a sample of the issues I see in CNV sections.
1. Lack of CNV and Structural Variant Detection (Lines: 137–138, 597 etc.)
The authors explicitly state that their pipeline does not call CNVs or large structural variants. This is a major limitation for a newborn screening program, as many actionable disorders are mediated by CNVs. While the discussion mentions WGS as a future possibility, the omission of any CNV detection from the current panel-based workflow is insufficiently justified. What would be the consequence for different genes and conditions? Would that be OK say in DMD gene majority of events are CNV events, then that would be hard to screen by a targeted panel? Many tools are used with targeted panels like CNVkit etc. why weren't they considered? I would like the following: Quantify the number of conditions in the BabyDetect panel that are primarily or frequently caused by CNVs or structural variants? Discuss why existing tools such as CNVkit were not employed, even in a limited fashion? Provide a roadmap for future CNV detection integration, and assess the potential false negative (FN) burden this omission introduces. Without that your not doing justice to your own work and perpetuating the concept that WGS may be the way to go or the technology is not ready. If on the other hand you say this is a starting point and we will do this in the following way over time or we suggest this would be the roadmap that will be understood by readers.
Line-Specific and Minor Comments: Below are a sampling of my observations.
-
Line 20: Consider summarizing BabyDetect’s design in the abstract in more accessible language and contrasting it more clearly with traditional biochemical NBS. "Targeted NGS" is technical—consider terms like “gene panel sequencing.”
-
Line 52: Sentence is awkward: “NBS is far to be implemented...” Suggest rephrasing to: “In many countries, comprehensive NBS programs are not yet standard components of neonatal care.”
-
Line 72: Clarify why “dedicated” filter paper cards were needed—this is not clear
-
Line 83: Gene names should be italicized throughout.
-
Line 94–105: The purpose of manual vs. automated DNA extraction is unclear. Summarize tradeoffs clearly—was automation adopted for throughput, quality, or cost reasons? It is hard to follow the rationale and it would be best to tell it like a story at least in the result section as to when we started we did this and then we did this to scale...etc.
-
Line 105: The rationale for redesigning the tNGS panel is underexplained. It reads generically. Clarify what limitations in panel-v1 prompted changes, and how panel-v2 improved performance. It seems like a copy paste.
-
Line 171 and related QC text: Many of the highly detailed QC metrics could be moved to supplementary material. As is, they distract from the key messages of performance and utility.
-
Lines 270–289: Consider adding a schematic of the variant interpretation decision tree to clarify the filtration and classification logic.
-
Gene Panel line 105: why two versions is unclear. Much of it is generic. Please provide a full list of genes (with added/removed flags) in the supplementary material along with inclusion/exclusion rationale.
Good luck

Author Response
Dear Reviewer,
We would like to thank you for reviewing our manuscript and for your constructive and insightful comments.
We will address here the questions you have kindly raised.
We thank the reviewer for raising these points. First, we acknowledge that two different numbers were reported in the manuscript in relation to BabyDetect samples, and we apologize for the lack of clarity. The number of samples analyzed and included in this study (validation and screened newborn samples) is over 5,900 (Line 26). In the Discussion, we had additionally mentioned the total number of newborns screened during the same period under the BabyDetect project (6,500; Line 622). However, as those additional samples were not part of the current analysis, we have removed that statement to avoid confusion for the reader.
With respect to the methodological aspects raised, our detailed responses are provided below:
- Reviewer-1 comment: Lack of CNV and Structural Variant Detection (Lines: 137–138, 597 etc.)
We thank the reviewer for this insightful comment. We fully agree that the detection of CNVs and other structural variants is an essential component of comprehensive genetic screening, as such variants can indeed be pathogenic and disease-causing. Several well-established methods exist for CNV detection from WES or targeted panel data, and some of these are already implemented in our laboratory.
In this first part of the study, however, we deliberately focused on small variant detection (SNVs and small indels). We plan to report CNV detection in a subsequent study, after conducting a thorough evaluation of detection performance, which requires a sufficient number of positive control samples for validation, as well as careful assessment of the pathogenicity of the identified CNVs.
In the revised manuscript, we have (i) clarified that CNV/SV detection is crucial for a comprehensive genetic-based newborn screening program, (ii) highlighted this importance by including illustrative examples and prevalence estimates of pathogenic CNVs from the literature, and (iii) outlined our roadmap for implementing CNV detection in future work.
The text has been edited as follows:
Line 70-75:
« In this first implementation, no copy-number or structural variant (CNV/SV) analysis was performed due to lack of sufficient positive controls for validation. The detection of CNVs and other SVs is an essential component of comprehensive genetic screening, as such variants can indeed be pathogenic and disease-causing. Here, we present a pragmatic plan to enable CNV detection and analysis in future releases while preserving screening-grade performance ».
Line 152-154:
« The workflow does not call copy-number variants (CNVs), large deletions, mosaicism, or other structural variants; no CNV analysis was performed in this study due to insufficient positive controls for validation ».
Line 562-602:
« A key limitation of the present manuscript is the absence of copy-number and structural variant analysis. We intentionally restricted scope to SNVs and short indels to maintain screening-grade positive predictive value (PPV), batch-to-batch stability, and turnaround time, given the lack of sufficient positive controls for pipeline validation. Indeed, SNVs represent the predominant cause of genetic disease, accounting for an estimated 85% of known pathogenic mutations [27, 28]. In comparison, CNVs and other SVs contribute to approximately 10–15% of genetic disorders [29]. CNVs, defined as gains or losses of DNA segments, can markedly alter gene dosage and function. Although CNVs span 12–16% of the human genome, only a subset are considered rare and clinically relevant [30]. These rare CNVs account for approximately 10% of SVs implicated in rare diseases [31]. However, not all rare pediatric disorders are equally enriched for pathogenic CNVs. Syndromic neurodevelopmental disorders (e.g., intellectual disability/global developmental delay with congenital anomalies or epilepsy) often exhibit high CNV yields (~20–30%), though SNVs and small indels contribute majorly to disease [32, 33]. Moreover, certain disorders, such as Duchenne muscular dystrophy or Angelman syndrome, are CNV-driven by definition, with CNVs representing the primary pathogenic mechanism (~70–80%). As such CNV-driven conditions were excluded from our screening gene list, based on our eligibility criteria, we have 17% of genes (68 out of 405 genes) that contain reported CNVs and other SVs, and in most of them the estimated contribution of pathogenic CNVs/SVs is approximately 2-5%.
As a prerequisite to routine deployment, we will conduct a staged validation using confirmed CNV-positive samples that reflect the diversity of events likely to occur in our assay: (i) single-exon versus multi-exon deletions/duplications; (ii) small versus large events in terms of the number of captured targets; (iii) loci with high homology or pseudogene interference; (iv) GC-rich and GC-poor segments; and (v) exons at capture boundaries. Acceptance criteria will include per-gene/region sensitivity estimates, PPV compatible with population screening, a stable per-sample call burden, and inter-run reproducibility. Upon meeting these criteria, CNV analysis will be activated in routine, with MLPA or qPCR confirmation required for any putative screen-positive sample prior to clinical reporting.
To mitigate the current scarcity of confirmed positives, we will also run a prospective, off-line “shadow” CNV evaluation across all incoming samples under a research protocol separate from routine screening and reporting. This will not affect turnaround time or clinical outputs. Candidates will be restricted to high-confidence patterns (e.g., multi-exon events in well-covered genes, consistent signal across QC checks with acceptable sample-to-reference correlation and a non-inflated per-sample CNV burden). Any candidate meeting predefined triage criteria will undergo orthogonal testing on retained material (MLPA/qPCR). Confirmed events will be used to refine thresholds, estimate gene-level sensitivity/PPV, and lock a sufficiently large panel-of-normals for robust normalization. Once screening-grade performance is demonstrated, CNV analysis will transition from shadow evaluation to routine initially on a CNV-ready subset of genes».
Line 778-782:
«While CNV/SV analyses were beyond the scope of the present work, we have outlined a staged plan for their integration in future studies: accrue positive controls for validation, establish a robust normalization cohort, and activate CNV calling with mandatory orthogonal confirmation once screening-grade acceptance criteria are satisfied».
- A second methodological gap, “limited indel precision” was indicated by the Reviewer-1 as a limitation for the manuscript.
We thank the reviewer for this valuable comment. We acknowledge that indel detection is inherently challenging. Our Humanomics pipeline supports the identification of small indels (1–15 bp) located within exons or at intron–exon boundaries (~50 bp flanking regions). To evaluate performance, we assessed sensitivity and precision using a GIAB sample.
For sensitivity (reflecting the negative predictive value and false-negative rate), we estimated ~80% for indels in both panel-v1 and panel-v2. Importantly, in our screened cohort we did not observe technical false negatives. Only one clinical false negative was identified (nonsense c.1030C>T homozygous variant in TJP2; F. Boemer, 2025). This variant was present in the raw sequencing data but was not included in our managed variant list at that time, and therefore excluded by the decision tree, due to lack of available knowledge on the variant and not because of technical limitations.
Regarding precision (reflecting the positive predictive value and false-positive rate), we estimated 59% for panel-v1 and 64% for panel-v2. Within our screened cohort, we observed one clinical false positive (AGXT gene: c.33dupC and c.332G>A; F. Boemer, 2025). Both pathogenic variants were in cis and inherited from an asymptomatic mother. Notably, this false positive was not due to technical limitations.
We recognize that technical false positives could increase interpretation burden given the relatively low indel precision. To further address this, we analyzed false-positive indels in a GIAB sample and provided results in the supplementary materials (Table S6). In addition, we included a new section in the revised manuscript (Lines 538–546). Specifically, we identified and investigated 51 indel variants from GIAB-2 sample captured with panel-v2 (Table S6). Most were located at exon–intron boundaries; a few were within the same gene and only a few were annotated in databases. The majority of these variants had low allele frequencies and/or were classified as benign with no clinical relevance for interpretation, and thus systematically excluded by our variant filtering tree. For this reason, we accept the relatively low precision values for indels.
The text has been edited as follows:
Line 538-546 - « It is important to indicate that, to further analyze the identified low precision of indels, we have investigated false-positive indels in a GIAB sample and provided results in the supplementary materials (Table S6). Specifically, we identified and investigated 51 indel variants from HG002-NA24385 reference sample captured with panel-v2. Most variants were located at exon–intron boundaries; a few were within the same gene and only a few were annotated in databases. The majority of these variants had low allele frequencies and/or were classified as benign with no clinical relevance for interpretation, and thus systematically excluded by our variant filtering tree. For this reason, we accept the relatively low precision values for indels. »
- Another methodological gap, such as “unphased variant calls”. was indicated by Reviewer-1 as a limitation for the manuscript.
We thank the reviewer for highlighting the issue of unphased variant calls as an additional methodological limitation. We agree that this is an inherent limitation of short-read sequencing, which does not allow phasing of variants without the availability of parental samples. This constraint may increase the false-positive rate and hinder the accurate determination of true compound heterozygotes in recessive conditions, as well as haplotypes versus heterozygotes in X-linked conditions. Long-read sequencing offers clear advantages in this regard and can overcome this limitation.
In the revised manuscript, we have added a statement in the Discussion section to explicitly acknowledge this point.
Line 646-656 - « It is important to note that although second-generation short-read sequencing is efficient and cost-effective, it lacks capacity to phase pathogenic variants without parental samples to assess segregation (cis or trans). The need to request parental samples can prolong the diagnostic process and cause anxiety for families while results are pending. Third-generation long-read sequencing platforms offer a solution to this challenge. A major advantage of long-read sequencing is its ability to provide phasing information directly from a single individual, linking variants located on the same DNA strand. This approach removes the requirement for parental DNA to determine variant inheritance in recessive conditions, thereby reducing diagnostic delays, alleviating parental anxiety, and streamlining laboratory workflows. Moreover, long-read sequencing enables more accurate detection of CNVs and other structural variants. »
- To improve transparency and data sharing, the full list of analyzed genes is now provided in the Supplementary Materials (Table S3), and the decision tree for variant interpretation is outlined in Figure 2.
- Reviewer-1 Line-Specific and Minor Comments: Below are a sampling of my observations.
- Line 20: Consider summarizing BabyDetect’s design in the abstract in more accessible language and contrasting it more clearly with traditional biochemical NBS. "Targeted NGS" is technical—consider terms like “gene panel sequencing.”
The sentences in the abstract have been modified as follows:
Line 20-22 - «The BabyDetect study used gene panel sequencing to expand NBS to treatable conditions not covered by conventional biochemical screening».
Line 29-31 - «Our findings demonstrate that gene panel sequencing-based NBS is feasible, accurate, and scalable, addressing critical gaps in current screening programs».
- Line 52: Sentence is awkward: “NBS is far to be implemented...” Suggest rephrasing to: “In many countries, comprehensive NBS programs are not yet standard components of neonatal care.”
The sentence in the text has been modified as follows:
Line 53-54 - « In many countries, comprehensive NBS programs have not yet been established as standard practices in neonatal care ».
- Line 72: Clarify why “dedicated” filter paper cards were needed—this is not clear
The following precision has been added in the text as follows:
Line 79-81 - « These cards were used to keep research samples separate from routine NBS workflow and to streamline logistics and traceability ».
- Line 83: Gene names should be italicized throughout.
Thank you for pointing out this oversight. Gene names have been modified accordingly lines 91, and 408 in the text and in supplementary Table S5 and Table S6.
- Line 94–105: The purpose of manual vs. automated DNA extraction is unclear. Summarize tradeoffs clearly—was automation adopted for throughput, quality, or cost reasons? It is hard to follow the rationale and it would be best to tell it like a story at least in the result section as to when we started we did this and then we did this to scale...etc.
We thank the reviewer for this helpful comment. To clarify the rationale and trade-offs between manual and automated DNA extraction, we have revised the Materials and Methods section. We now describe the progression from manual to automated extraction, highlighting the reasons for this transition, namely scalability and improved turnaround time.
The revised text reads as follows:
Lines 101-112 - Materials and Methods « DNA extraction » section:
« DNA was extracted from DBS and whole blood manually using QIAamp DNA Investigator Kit (Qiagen, USA) according to manufacturer’s instructions with modifications for plate-based extraction. Manual extraction was used for initial validation samples as well as during the first year of screening.
After one year of screening, in order to ensure scalability of the population-based NBS and to improve turnaround time (TAT), automated extraction was implemented using the QIAsymphony SP instrument with the QIAsymphony DNA Investigator Kit (Qiagen). Both manual and automated methods were validated and compared.
DNA yield was quantified using a Qubit fluorometer (Thermo Fisher Scientific), and DNA quality and fragment size were assessed by agarose gel electrophoresis and Agilent fragment analysis (Figure S1).
Results « Workflow » section:
The sentence was modified as follows:
Lines 239-245 - «To optimize DNA yield, we initially incorporated an overnight lysis of DBS punches at 30 °C, followed by manual DNA extraction using the Qiagen column-based protocol. This approach was applied to validation samples and one-year screening samples. As the study progressed and the need for scalability and shorter TAT became evident, we validated and transitioned to automated extraction using the QIAsymphony platform. This automated method was subsequently adopted for the remaining study samples».
- Line 105: The rationale for redesigning the tNGS panel is underexplained. It reads generically. Clarify what limitations in panel-v1 prompted changes, and how panel-v2 improved performance. It seems like a copy paste.
- Gene Panel line 105: why two versions is unclear. Much of it is generic. Please provide a full list of genes (with added/removed flags) in the supplementary material along with inclusion/exclusion rationale.
We thank the reviewer for these valuable comments. We agree that the rationale for redesigning the tNGS panel and the distinction between panel-v1 and panel-v2 required further clarification. In the revised manuscript, we have expanded the description of the redesign and provided supporting materials.
- Materials and Methods : We now explain that the primary technical motivation for panel-v2 was to improve on-target capture efficiency.
Sentence was added as follows :
Line 131-136 - « The primary technical rationale for redesigning panel-v2 was to improve on-target capture efficiency. In panel-v1, intronic regions as well as 3′ and 5′ UTRs from 20 genes had been included in the target of interest (TOI), but these noncoding regions were excluded in panel-v2 to enhance performance. In addition, the redesign enabled the incorporation of newly curated genes of interest identified during a second round of gene curation (Table S3) ».
- Supplementary Table S3:We have created a comprehensive gene list that specifies all included and excluded genes, with annotations indicating which genes were added or removed between versions. The legend to Table S3 also outlines the criteria used for inclusion and exclusion.
Additionally, we have added the gene inclusion criteria in the text as follows:
Line 115-118 - « Gene inclusion criteria were considered before selecting genes of interest: early onset of disease (before age of 5), severe disease/ disability if not treated, available treatment, benefit from treatment, pathogenic mutation, phenotype-genotype correlation and standard of care (Table S3)».
Throughout the text, Table S3 is referred in lines, 118, 121, 136, 510.
- Results section (Target of interest: improving performance and clinical impact » Line 508): We have a dedicated section describing how the redesign of panel-v2 improved both technical performance and clinical yield.
- Line 171 and related QC text: Many of the highly detailed QC metrics could be moved to supplementary material. As is, they distract from the key messages of performance and utility.
The quality control (QC) parameters detailed in the Materials and Methods under the section «Definition and selection of quality metrics» (line 187) were removed from the text and placed in Supplementary materials and an additional supplementary table (Table S4) was created. Table S4 is referred in the text on line 191.
- Lines 270–289: Consider adding a schematic of the variant interpretation decision tree to clarify the filtration and classification logic.
The schematic of variant interpretation decision tree has been created and placed in the text as Figure 2 (line 290). Figure 2 is referred in the text on line 271.
In addition to the changes already described, we have made the following further modifications:
- Additional articles have been incorporated into the references (Refs. 27–33).
- Table 1 has been moved from the main text to the Supplementary Materials as Table S5 and is now referred to in the text under Lines 306 and 314.
- Table S6 is prepared for Supplementary materials illustrating the analysis of False positive indel variants from GIAB sample.
- The manuscript has also undergone an English language review to improve clarity and readability.
- The figures are edited using the Figure editing service of MDPI
- Tables are revised according to the guidelines of MDPI
We hope that the explanations provided in the cover letter and the modifications implemented in the text will be useful to our readers and contribute to improving the quality of the manuscript.

Reviewer 2 Report
Comments and Suggestions for Authors
The manuscript is well written and clearly structured, demonstrating a strong grasp of the subject matter.I only found the following statement confusing. Could you please confirm that it conveys what you intended? Other than that, the manuscript is ready for publication.
Comments:
line 603 - "Our developed filtering strategy and conservative approach in reporting only known pathogenic and likely pathogenic variants allowed us to identify 1.8% of positive cases, with 0.8% not identified by conventional screening"
This must be an error. This statement means that you only identified 1.8% of positive cases, which leaves 98.2% of positive cases as unidentified.
Do you mean that 1.8% of the total cases were identified as positive?
Author Response
Comment of the reviewer: The manuscript is well written and clearly structured, demonstrating a strong grasp of the subject matter.I only found the following statement confusing. Could you please confirm that it conveys what you intended? Other than that, the manuscript is ready for publication.
Comments:
line 603 - "Our developed filtering strategy and conservative approach in reporting only known pathogenic and likely pathogenic variants allowed us to identify 1.8% of positive cases, with 0.8% not identified by conventional screening"
This must be an error. This statement means that you only identified 1.8% of positive cases, which leaves 98.2% of positive cases as unidentified.
Do you mean that 1.8% of the total cases were identified as positive?
Dear Reviewer,
We sincerely thank you for pointing out this error and for careful reading of our manuscript. The reviewer is correct that the original statement was misleading. Our intended meaning was that 1.8% of the total screened cases were identified as positive (71 positives out of 3,847), not that only 1.8% of positive cases were detected.
We have corrected the sentence accordingly:
Lines 660–662 - "Our developed filtering strategy and conservative approach, which reported only known pathogenic and likely pathogenic variants, enabled us to identify 1.8% of screened cases as positives, with 0.8% not identified by conventional screening [19]. "

Round 2
Reviewer 1 Report
Comments and Suggestions for Authors
Edits look fine and is acceptable.
Author Response
Dear Reviewer,
Thanks a lot for your kind suggestions and comments, which improved our manuscript to be accepted for publication.
Kindest regards,
Kristine Hovhannesyan and co-authors